# Where Do I Allocate My Urban Allotment Gardens? Development of a Site Selection Tool for Three Cities in Benin

**B. G. J. S. Sonneveld** [1,2,*], **M. D. Houessou** [1,3,4], **G. J. M. van den Boom** [2] and **A. Aoudji** [3]

1 Athena Institute, Vrije Universiteit Amsterdam, De Boelelaan 1085, 1081 HV Amsterdam, The Netherlands; d.houessou@aced-benin.org

2 Amsterdam Centre for World Food Studies, School of Business and Economics, Vrije Universiteit Amsterdam, De Boelelaan 1085, 1081 HV Amsterdam, The Netherlands; g.j.m.vanden.boom@vu.nl

3 Laboratoire d'Etude sur la Pauvreté et la Performance de l'Agriculture (LEPPA), Faculty of Agricultural Sciences, University of Abomey-Calavi, R.P. Cotonou 03 BP 2819, Benin; augustin.aoudji@fsa.uac.bj

4 Centre d'Actions pour l'Environnement et le Développement Durable, Abomey-Calavi BP 660, Benin

\* Correspondence: b.g.j.s.sonneveld@vu.nl; Tel.: +31-(0)-616-782-007

**Abstract:** In the context of rapid urbanization, poorer residents in cities across low- and middle-income countries increasingly experience food and nutrition deficiencies. The United Nations has highlighted urban agriculture (UA) as a viable solution to food insecurity, by empowering the urban poor to produce their own fresh foods and make some profit from surplus production. Despite its potential role in reducing poverty and food insecurity, there appears to be little political will to support urban agriculture. This is seen in unclear political mandates that are sustained by information gaps on selection criteria for UA sites. The research reported here addresses this issue in the form of a decision-making support tool that assesses the suitability of cadastral units and informal plots for allotment gardens in urban and peri-urban areas. The tool was developed and tested for three rapidly expanding cities in Benin, a low-income country in West Africa, based on an ordered logit model that relates a set of 300 expert assessments on site suitability to georeferenced information on biophysical and socio-economic characteristics. Soil, land use, groundwater depth, vicinity to market and women's safety were significant factors in the assessment. Scaled up across all cadastral units and informal sites, the tool generated detailed baseline maps on site suitability and availability of areas. Its capacity to support policymakers in selecting appropriate sites comes to the fore by reporting changes in site suitability under scenarios of improved soil fertility and enhanced safety for women.

**Keywords:** urban planning; GIS; urban agriculture; cities; Benin; Africa

## 1. Introduction

It has been argued that urban agriculture (UA) in low- and middle-income countries contributes to food security and poverty reduction by enabling people to produce fresh and nutritious food and reduce their food expenditure [1]. There have been examples from Gabon [2], Democratic Republic of Congo [3] and Malaysia [4]. UA may also contribute to more sustainable and resilient urban communities in view of its pivotal role in circular-economy strategies at the city level, while contributing to restoring natural cycles and providing environmental services [5]. Indeed, UA offers an opportunity to process a large proportion of urban waste into compost, an organic fertilizer for crop production. Ecologically, UA also contributes to reducing the effect of "heat islands" by creating buffer zones, while city residents appreciate the beauty of green areas. However, land pressure in urban and peri-urban areas makes it hard to expand UA initiatives because land is primarily dedicated to building housing, industries, and other



infrastructures. As a result, and despite its multiple benefits, UA remains marginalized and rural and urban planning protocols are lacking.

Faced with mounting demand for land for newcomers, wealthy households, and large companies, it is increasingly difficult to allocate sites for small-scale producers in and around cities. For instance, a recent study in Tamale, Ghana, showed that the most serious threat to farmers posed by urbanization is the reduction in prime agricultural land, with the resulting negative effects for agricultural production, food security and the standard of living [6]. An important factor is the virtual absence of rural and urban planning schemes to allocate land to UA. There are two main reasons for this. First, experience shows the lack of a clear mandate for national and local authorities to promote and implement urban gardens. Second, authorities often lack a structure to appreciate the suitability of sites for UA, making it difficult to develop a coherent policy on urban land use and development.

Benin is a typical case in point due to the place of UA in its cities' food systems. In Benin, UA is of paramount importance for food security and poverty reduction, especially in cities whose populations have grown rapidly in recent years [7,8]. For instance, while the national population growth rate between 2002 and 2013 was 3.5%, it was over 6% in newer cities such as Abomey-Calavi, Ouidah, Seme-Kpodji [9]. This was caused by internal growth combined with a rural exodus and the establishment of newer cities. Though the proportion of food-insecure people in Benin's towns and cities remained at a stable 8% in the period 2012–2017, the absolute number increased from around 332,000 to 449,000. Moreover, many poor urban households in Benin cut back on the more expensive nutritious foods [10–12] and replaced them with cheaper, highly calorific foods, thus reducing dietary diversity and micronutrient consumption, and increasing the risk of adverse health consequences and malnutrition [13,14]. Hence, urban administrations in Benin face many challenges in curbing the threat of food insecurity and safeguarding sustainable access to fresh and healthy foods for poor urban dwellers [15]. Participation in UA is a viable and sustainable answer to the problem of food insecurity and also brings multiple benefits. First, UA enables people living in poverty to address the issue of food insecurity and cultivate the crops they need, enriching their diet and earning some income with the surplus produce [16]. Second, UA can play an important role in recycling organic waste as compost, to enrich the soil. Third, UA provides green areas in the cities that have a cooling effect, and enhance the urban environment [17,18]. Despite these positive impacts, expanding urban agriculture is not occurring at the anticipated pace. A thorough analysis shows that expansion is constrained by a number of challenges that are underpinned by the lack of knowledge on the availability and suitability of possible areas in and around the cities [19,20]. The allocation of land for UA is particularly pressing for local governments in Benin because since the inception of the decentralization process in 2003, municipalities are mandated to organize land-use planning. However, experience shows that there is a persistent lack of information on which to design land-use schemes [21], while specific allocation protocols for UA are mostly non-existent or not operational [22]. A main reason for the lack of support tools for decisions about the allocation of land for UA is that the question of what makes a site suitable entails a multifactorial and integrated approach that takes account of the agronomic, geographic, socio-economic and gender-related requirements. Hence, calls for clear protocols to assign land to UA are justified and should combine knowledge from various scientific disciplines with practitioners' experience to generate evidence-based knowledge that support policies in the planning of UA.

Our research addresses this issue and adopts a transdisciplinary approach that integrates perspectives from academic knowledge and experts' judgements to provide a field-informed decision-support tool. Therefore, our research departs from a spatial multi-criteria-decision-making procedure to assess land suitability. Spatial decision problems typically involve a large set of feasible alternatives and multiple, conflicting, and incommensurate evaluation criteria [23]. The alternatives are often evaluated by a number

of individuals (decision-makers, managers, stakeholders, interest groups) who are typically characterized by unique preferences with respect to the relative importance of criteria on the basis of which the alternatives are evaluated. Accordingly, many spatial decision problems give rise to the GIS-based multicriteria decision analysis (GIS-MCDA). On the one hand, GIS techniques and procedures have an important role to play in analyzing decision problems as GIS is often recognized "as a decision support system involving the integration of spatially referenced data in a problem-solving environment" [24]. On the other hand, MCDA provides a rich collection of techniques and procedures for structuring decision problems, and designing, evaluating and prioritizing alternative decisions. Therefore, GIS-MCDA can be thought of as a process that transforms and combines geographical data and value judgments (the decision-maker's preferences) to obtain information for decision making [23]; the analytical hierarchy process (AHP) being the widely used method for MCDA. The AHP method determines the weight of importance of different factors influencing land suitability based on either the existing literature or experts' opinion [25]; with a judgement bias which may influence the outcome. Hence, Ustaoglu and Aydınoglu [26] suggest the application of other robust suitability assessments methods such as MCDA integrated with logistic regression and a GIS approach. Our methodology innovates and contributes to the body of knowledge in two ways. First, data from various disciplines and GIS-based data formats are integrated into an analytical framework for weighted regression [27–30]. Then, the approach makes it possible for policymakers to use the tool to observe changes and choose among various options; a dynamic and instrumental feature that should facilitate policy uptake in urban planning in Benin and African countries more widely.

The paper is structured as follows. Section two describes the research methodology. Section three presents the results from data manipulation, estimation, and extrapolation. Section four discusses the results. Section five concludes and makes projections for future research and policy uptake.

## 2. Materials and Methods

The methodology followed three phases (Figure 1): a compilation of independent and dependent variables, the conversion of maps into vectors and the estimation of parameters, and the assessment of the suitability of unvisited sites for various options.

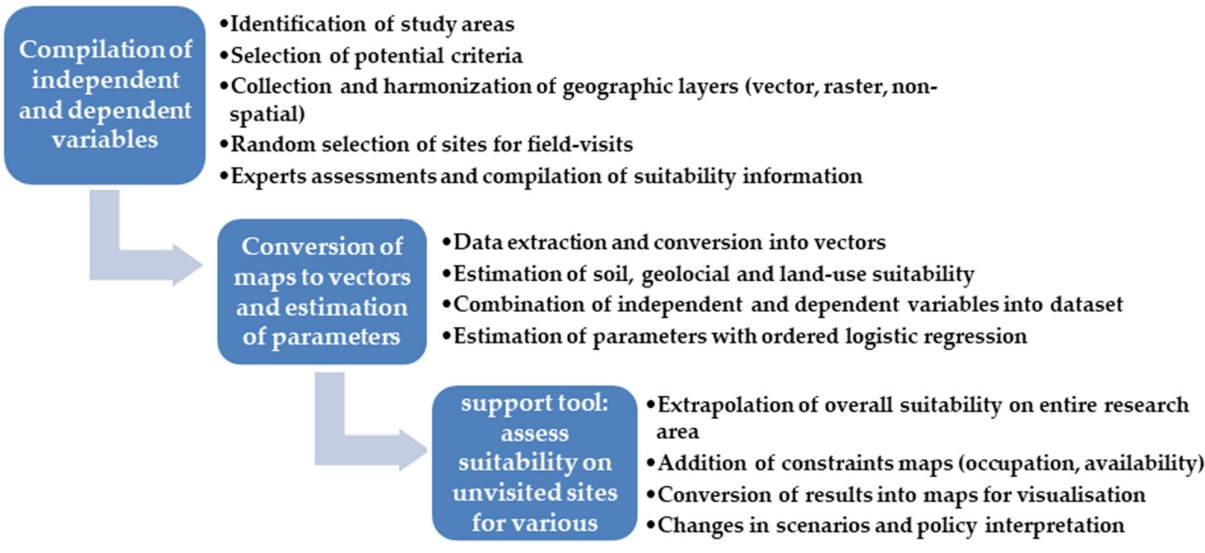

**Figure 1.** Methodological steps of the GIS-MCDA.

### 2.1. Study Areas

The research was conducted in three cities in the southern part of the republic of Benin: Abomey-Calavi, Cotonou, and Porto-Novo, that experienced a rapid growth. The exact number is unknown but the World Bank [31] shows that the share of urban population shifted from 38 to 48% from 2000 to 2019, which over the same period means an extra 3 million urban inhabitants. The three cities are currently the most densely populated urban areas in the country with 679,012, 656,358 and 264,320 citizens in 2013 for Cotonou, Abomey Calavi and Porto Novo respectively [9]. All three cities experience an increasing incidence of human poverty and thereby, of food insecurity [32,33] with limited social safety nets measures. Hence, the cities formed an interesting case study to conduct this research to identify open and suitable areas that may be useful to support the improvement of food security within urban areas.

### 2.2. Compilation of Independent and Dependent Variables

During a three-day science shop with thirteen participants a set of criteria was defined in relation to the suitability for allotments. The participants included UA experts from the ministry of agriculture, UA's practitioners, and researchers from soil, land, geographic, and social sciences. Thus, in a first attempt, a full list of independent variables was prepared and discussed in plenaries. The variables were selected based on their spatial availability and potential explanatory power to appraise site suitability for urban gardens. Most variables could be combined and resulted in a final list (Table 1). Then, we collected the corresponding geographic layers from various sources. Since the data were collected in different formats (raster, vector, and non-spatial datasets) and scales, we harmonized the data according to a standard georeference with a grid size of one arc-second (30 m), covering our research areas (the cities: Abomey-Calavi, Cotonou and Porto-Novo). The surface waters map collected was omitted because there were too many gaps in the data. There were no available data regarding safety for women, groundwater depth, soil and groundwater pollution, and markets. Hence, because of their importance, we proxied safety for women, distance to markets, and groundwater depth as follows: safety for women was estimated by computing the Euclidean distance between grids that are potentially suitable for UA and the nearest built-up area, derived from cadaster units. Distance to markets was estimated by computing the population density)per district [33], and groundwater depth was estimated by using the digital elevation model.

**Table 1.** Compilation of independent variables.

| Variables | Data | Measurements | Modalities | Sources of Data (assessed on 7 September 2020) |
|---|---|---|---|---|
| Cadaster units | Cadaster map | Availability levels | 1 = fully available<br>2 = partly available<br>3 = not available | Cadasters https://www.cadastre.bj/ |
| Soil suitability | Soil types map | Suitability levels | 1 = non-suitable<br>2 = slightly suitable<br>3 = moderately suitable<br>4 = suitable<br>5 = very suitable | Centre National de Télédétection et de Suivi Ecologique (CENATEL) |
| Road accessibility | Road map | Euclidean distance between grids and nearest road | Distance | CENATEL |
| Land-use suitability | Land cover map | Suitability levels | 1 = non-suitable<br>2 = slightly suitable<br>3 = moderately suitable<br>4 = suitable<br>5 = very suitable | CENATEL |
| Groundwater depth | Groundwater depth map | Altitude | Distance to groundwater | Proxied in digital elevation model (30*30) |

|  |  |  |  | https://earthexplorer.usgs.gov/ |
|---|---|---|---|---|
| Safety for women | Safety map/thefts statistics | Number of thefts reported per district | Number of thefts reported per district | Proxied in distance to built-up area map |
| Length of growing period | Length of growing period map | Number of growing days per year | Number of growing days per year | http://www.gaez.iiasa.ac.at/ |
| Geological suitability | Geological map | Suitability levels | 1 = non-suitable<br>2 = slightly suitable<br>3 = moderately suitable<br>4 = suitable<br>5 = very suitable | CENATEL |
| Surface water availability | Surface water map | Distance to temporary and permanent water | Distance | CENATEL |
| Soil and groundwater pollution | Soil and groundwater pollution map | Suitability levels | 1 = non-suitable<br>2 = slightly suitable<br>3 = moderately suitable<br>4 = suitable<br>5 = very suitable | None |
| Distance to market | Markets map | Euclidean distance between grids and nearest market | Distance | Proxied in population density map |

We compiled the dependent variable on the suitability of sites for UA on the basis of assessments by five UA experts who visited and assessed 60 randomly selected sites in our research area (Figure 2, distribution of random points visited map). The experts were experienced in UA development and were selected from gardeners, practitioners, and researchers to form a diversified and complementary research team. The random selection followed three steps. First, we stratified the sharing of points proportionally to the area ) of each city (Table 2) [33]. Second, we computed in ESRI ARCGIS pro 1.2, a random selection of points with a minimum distance of 1km between two points. After that, we found the possibility of increasing the number of points of Cotonou to eight and of Porto-Novo to ten; the remaining 42 were attributed to Abomey-Calavi. Finally, we computed again the random selection based on this sharing and recorded and extracted the coordinates of the angles of selected sites by the GPS. The overall assessment of visited sites accounted for the following characteristics: soil fertility, climate, surface and groundwater water accessibility, distance to markets, distance to households, safety for women and current land use. Then, based on observations, experts assessed the sites as suitable (60.67%), moderately suitable (22.67%) or unsuitable (16.67%).

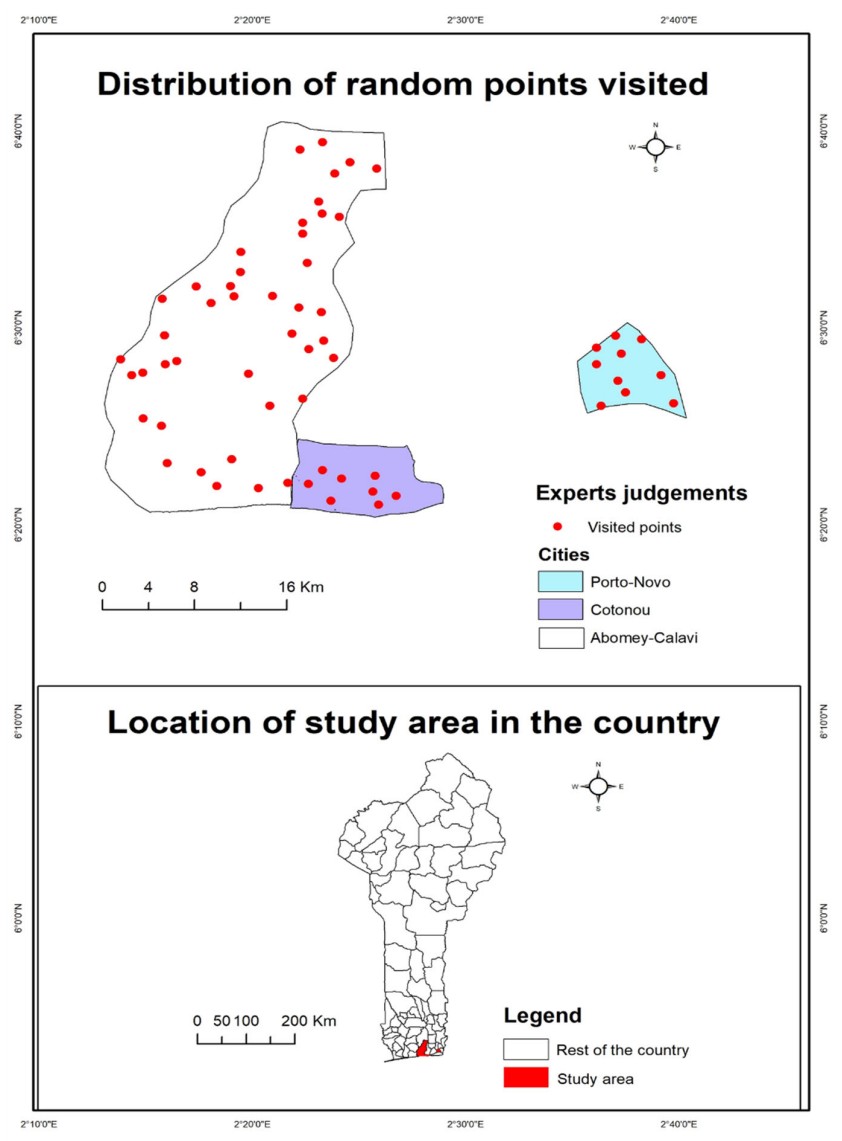

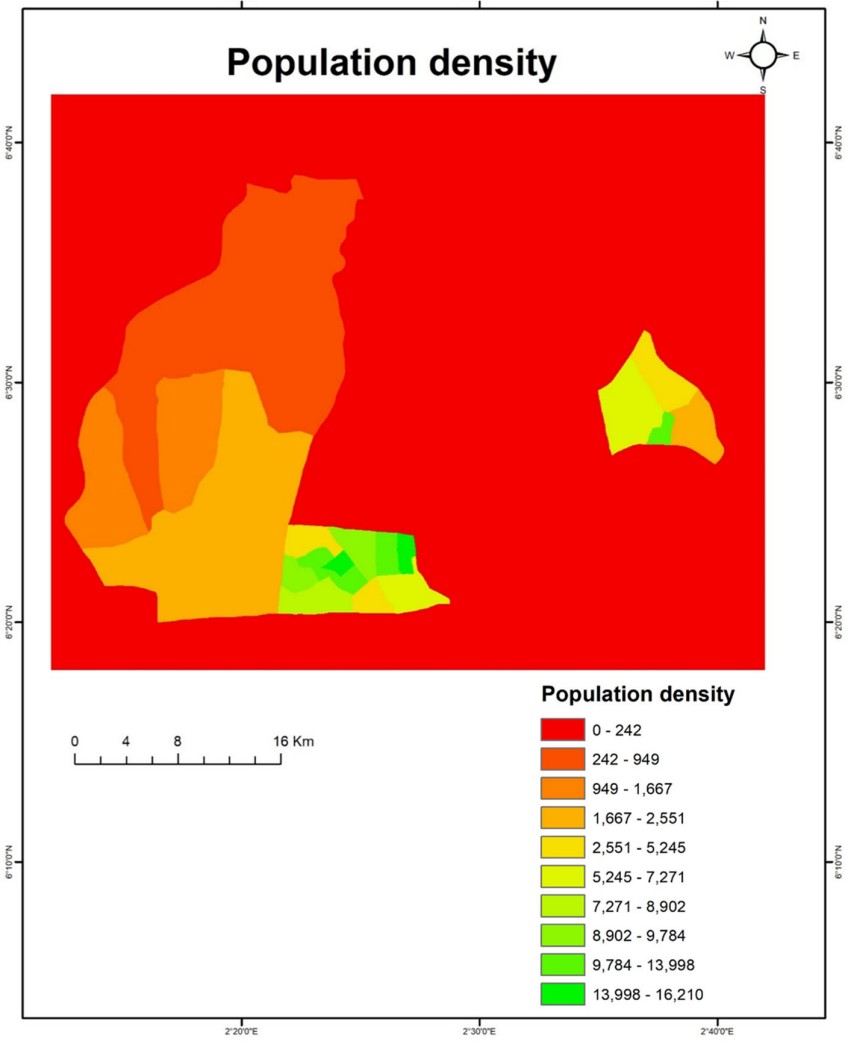

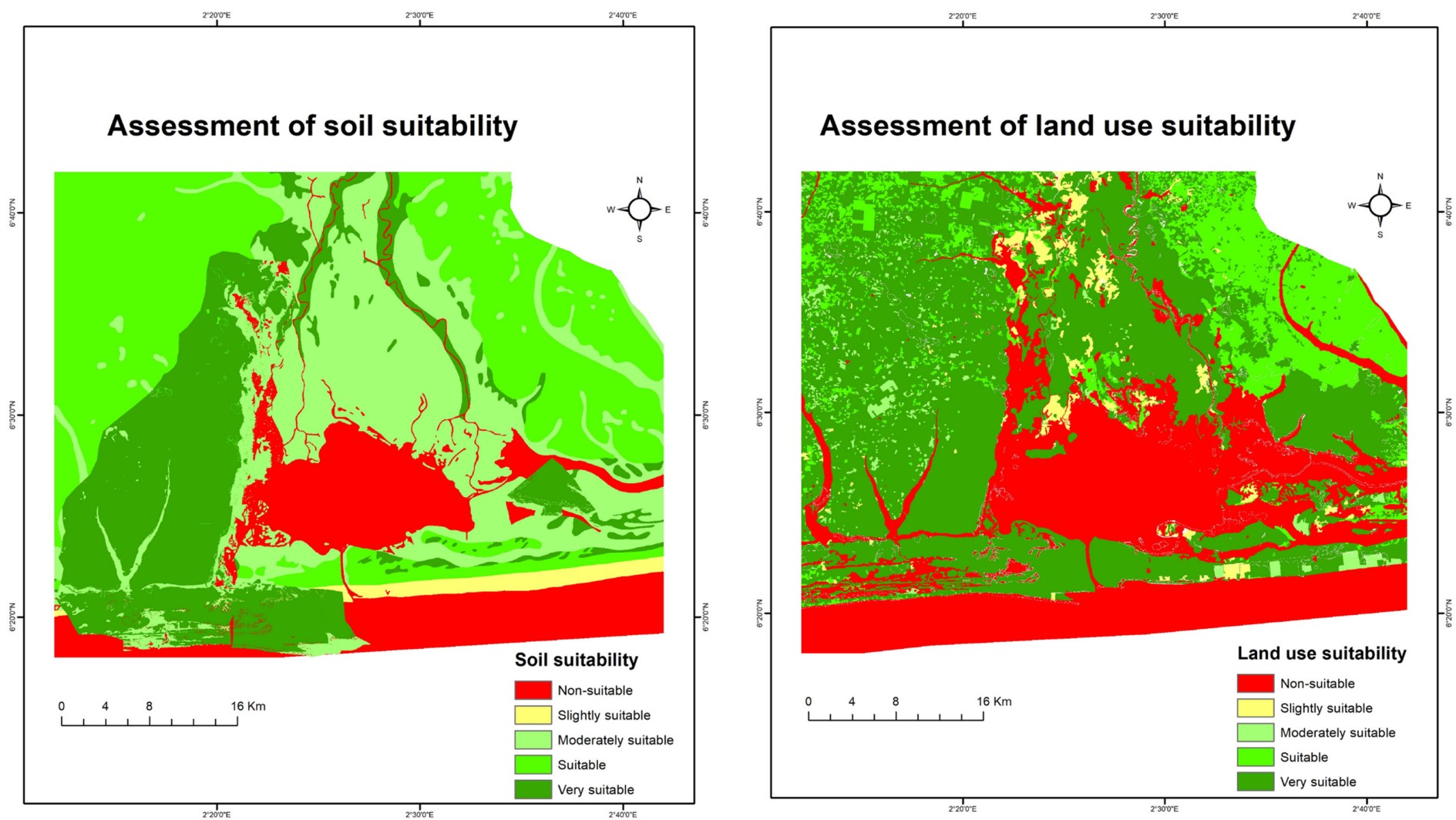

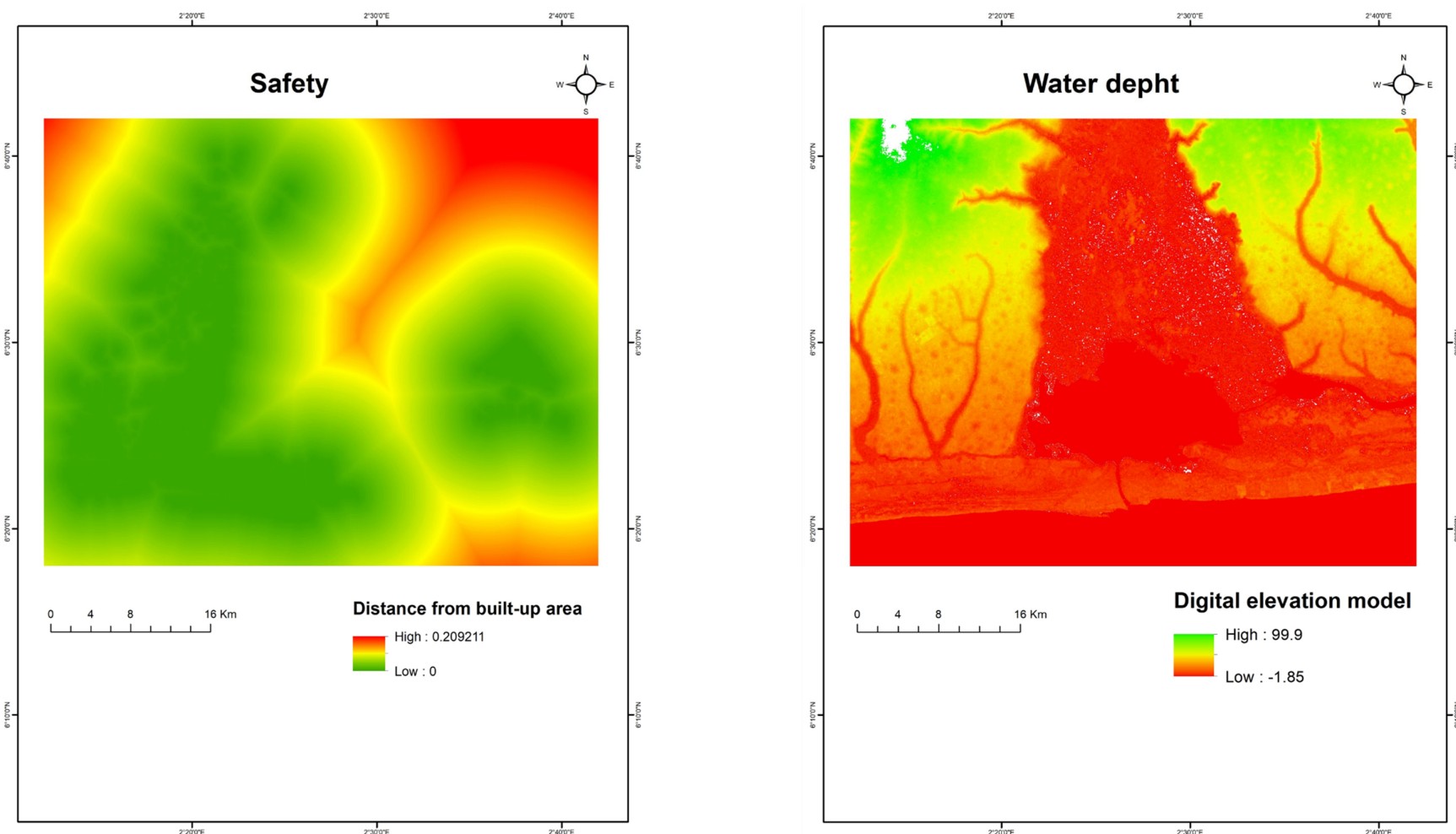

**Figure 2.** Classified maps of visited points and significant variables for site suitability.

**Table 2.** Share of visited points per city.

| City | Area (km²) | Share per Area | Number of Points Visited |
|---|---|---|---|
| Abomey-Calavi | 650 | 46 | 42 |
| Cotonou | 79 | 6 | 8 |
| Porto-Novo | 110 | 8 | 10 |
| Total | 839 | 60 | 60 |

*2.3. Conversion of Maps to Vectors and Estimation of Parameters*

Data relating to the independent variables were extracted from the collected maps (shapefiles and raster files) and converted into vectors using a dedicated software program (GIS2SAS) in SAS 9.4. The soil, geological and land-use suitability was estimated (see Table 3 and Table A1) using widely accepted international classifications [34,35]. The original names of the soil types were in French and it was not obvious to identify their exact and relevant correspondences in anglophone literature. Therefore, and for ease of references, the soil types were classified using their original French classification (see Table A1) that was translated in Table 3.

**Table 3.** Evaluation of soil, geological and land-use suitability.

| Items | Suitability Level |
|---|---|
| **Soil types** | |
| *Water body* | Unsuitable |
| *Hydromorphic soils* | Slightly suitable |
| *Hydromorphic vertisols* | Slightly suitable |
| *Leached tropical ferruginous soils without concretions* | Moderately suitable |
| *Hydromorphic leached tropical ferruginous soils* | Slightly suitable |
| *Low desaturated modal depleted ferrallitic soils* | Suitable |
| *Low desaturated hydromorphic ferrallitic Soils* | Moderately suitable |
| *Moderately organic humic to gley hydromorphic soils* | Moderately suitable |
| *Mineral or slightly humic hydromorphic soils with deep gley* | Moderately suitable |
| *Mineral or slightly humic hydromorphic soils with leached gley* | Suitable |
| *Hydromorphic soils with mineral or low humic content and pseudo-gley* | Very suitable |
| **Geology** | |
| *Water body* | Unsuitable |
| *Quaternary: aeolian and marine sands of the present and recent coastline, sandy–clay alluvium of the interior river valleys* | Moderately suitable |
| *Quaternary: clayey–sandy alluvium of the interior valleys of the rivers* | Very suitable |
| *Quaternary: terrace of 5 to 40 m very developed in clay and sand on the coastal facade* | Slightly suitable |
| *Mio-Pliocene terminal continental: lateritic red clay, variegated clay, sandy, black or colored clay, fine to coarse sand, sandstone* | Suitable |
| *Eocene and Paleocene terminal continental: blue-gray clay, quartz pebble bank, white fine sand, phosphate limestone* | Moderately suitable |
| *Upper Proterozoic Cambrian: mudstone, sils, fine sandstone, fine and medium quartzite, siltstone, jasper, shale* | Suitable |
| *Pan-African: syeno-monzonite, microsyenite granites, syntectonic calc-alkaline granites, charnockites, gness granitoides, nigmatic granites* | Very suitable |
| **Land cover** | |
| *Gallery forest* | Slightly suitable |
| *Dense forest* | Unsuitable |
| *Swamp forest* | Unsuitable |
| *Open forest and wooded savannah* | Slightly suitable |
| *Tree and shrub savannah* | Slightly suitable |
| *Plantation* | Moderately suitable |
| *Crops and fallow land* | Very suitable |
| *Crops and fallow land with palm trees* | Suitable |

| Rice cultivation | Slightly suitable |
|---|---|
| Mangrove | Unsuitable |
| Swamp | Unsuitable |
| Water body | Unsuitable |
| Agglomeration | Very suitable |
| Bare soil | Unsuitable |

This table was translated from the original French classification found in Table A1.

Then, dependent and dependent variables were combined into a dataset from which missing observations were deleted. This concerns some points located at the borders of the study area. Then, a stepwise ordered logistic regression was applied to explain the overall suitability of visited sites against the following explanatory variables: soil suitability, road accessibility, land-use suitability, geological suitability, groundwater depth, length of growing period, safety for women, population density. An ordered logistic regression is a method that estimates the probabilities of correct classifications of an event, with an ordinal ranking; the hit ratio (predicted values against observed values) was used to check the model accuracy on correct classifications [36]. Following Sonneveld et al. [37] we briefly explained the ordered logit model which represents the underlying process by

$$y_i = \beta y_i + \varepsilon_i$$

with the additive error terms being identically and independently distributed (iid) across observations, $\beta$ a vector of estimated parameters, $y_i$ the dependent variable, subscript i the observation number. What we observe is a site suitability class $z_i$ that represents ordered classes (1 = suitable; 2 = moderately suitable; 3 = non-suitable). The adjacent intervals of $y_i$ correspond with qualitative information $z_i$, as:

$$
\begin{aligned}
z_i = 1 &\quad \text{if} \quad y_i < \mu_1, \\
z_i = 2 &\quad \text{if} \quad \mu_1 \leq y_i < \mu_2, \\
z_i = n &\quad \text{if} \quad \mu_{n-1} \leq y_i.
\end{aligned}
$$

Using the maximum likelihood method, we estimated parameters $\beta$ and thresholds ($\mu_1$, ..., $\mu_{n-1}$) simultaneously with as driving force a maximization of correct classifications of site suitability classes assigned by experts. We calculated the probability (Pr) that $z_i = n$ by

$$\Pr(z_i = n) = \Pr(y_i \geq \mu_{n-1}) = \Pr(\epsilon_i \geq \mu_{n-1} - \beta x_i) = F(\beta x_i - \mu_{n-1}).$$

Disturbances $\varepsilon_i$ follow a logistic distribution that leads to a cumulative logistic transformation function $\Lambda$ that maps the admissible area of y, i.e., $(-\infty, \infty)$, to [0,1], with a first derivative that is always positive. For example, the likelihood function for the ordered logit model for n = 3 is given by

$$\ell(\beta, \mu_1, \mu_2) = \prod_{y_i=1} \Lambda(\mu_1 - \beta x_i) \prod_{y_i=2} \Lambda(\mu_2 - \beta x_i) - \Lambda(\mu_1 - \beta x_i) \prod_{y_i=3} \Lambda(\beta x_i - \mu_2),$$

where function $\ell$ is minimized with respect to the parameters $\beta$, $\mu_1$ and $\mu_2$.

### 2.4. Decision Support Tool: Assess Suitability on Unvisited Sites for Various Options

The estimated model was used to extrapolate the overall suitability for UA on the entire research area. Converting the results obtained in vector format into an * asc map we used dedicated software (SAS2GIS). The obtained overall suitability assessment for UA was combined with the availability of cadaster units (fully available (30%), partly available (22%), not available (49%) to generate further information for policymaking on UA development. Two policy intervention scenarios were then suggested to observe changes in overall suitability for urban agriculture. The first scenario improved the soil fertility to one level up and the second increased the safety for women by reducing the distance to a built-up area to a maximum range of 500 m. All the outcomes were visualized on maps (Figures 3–6) to facilitate policy interpretation.



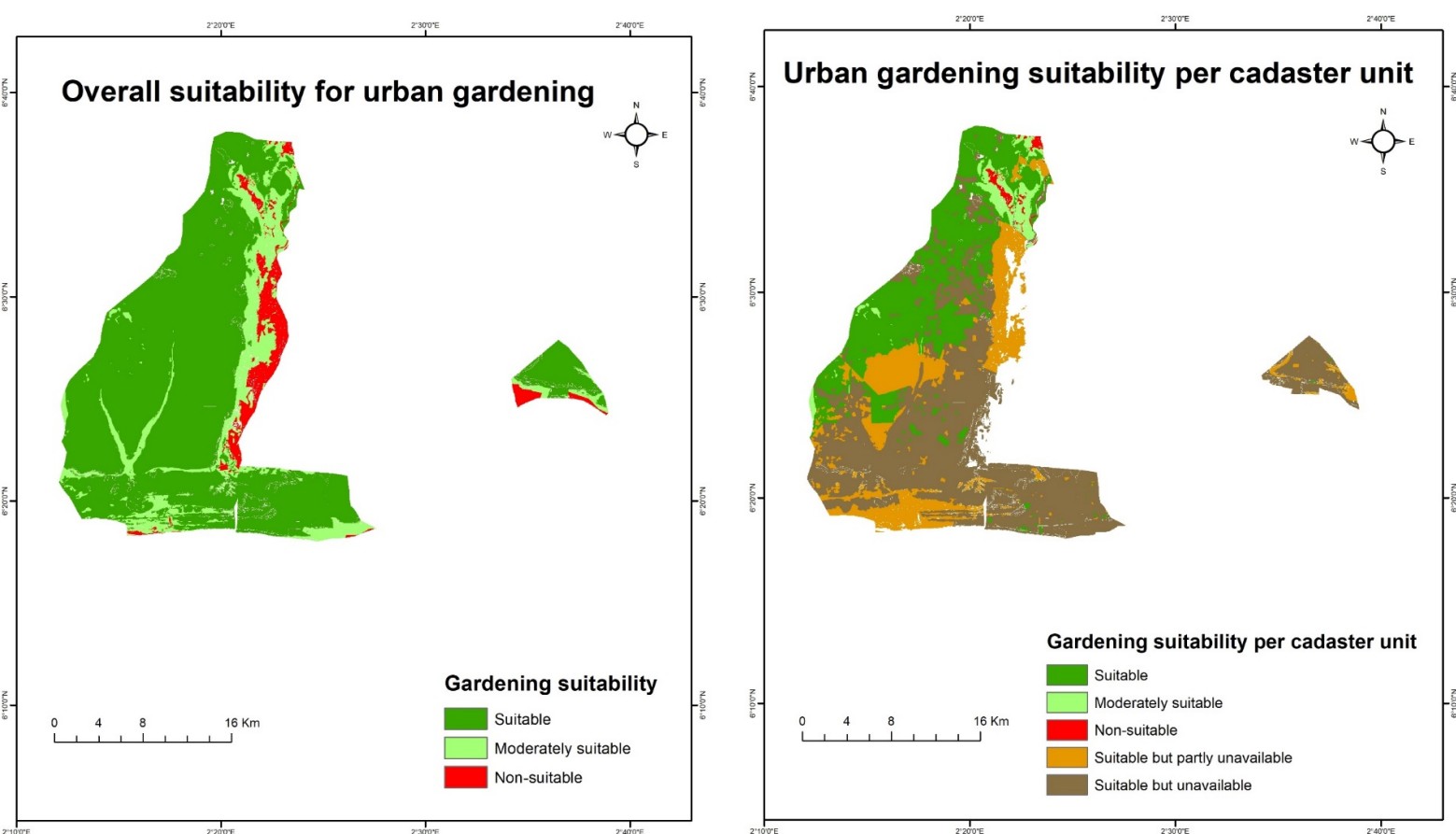

**Figure 3.** Classified maps of suitable areas for urban agriculture: baseline scenario.

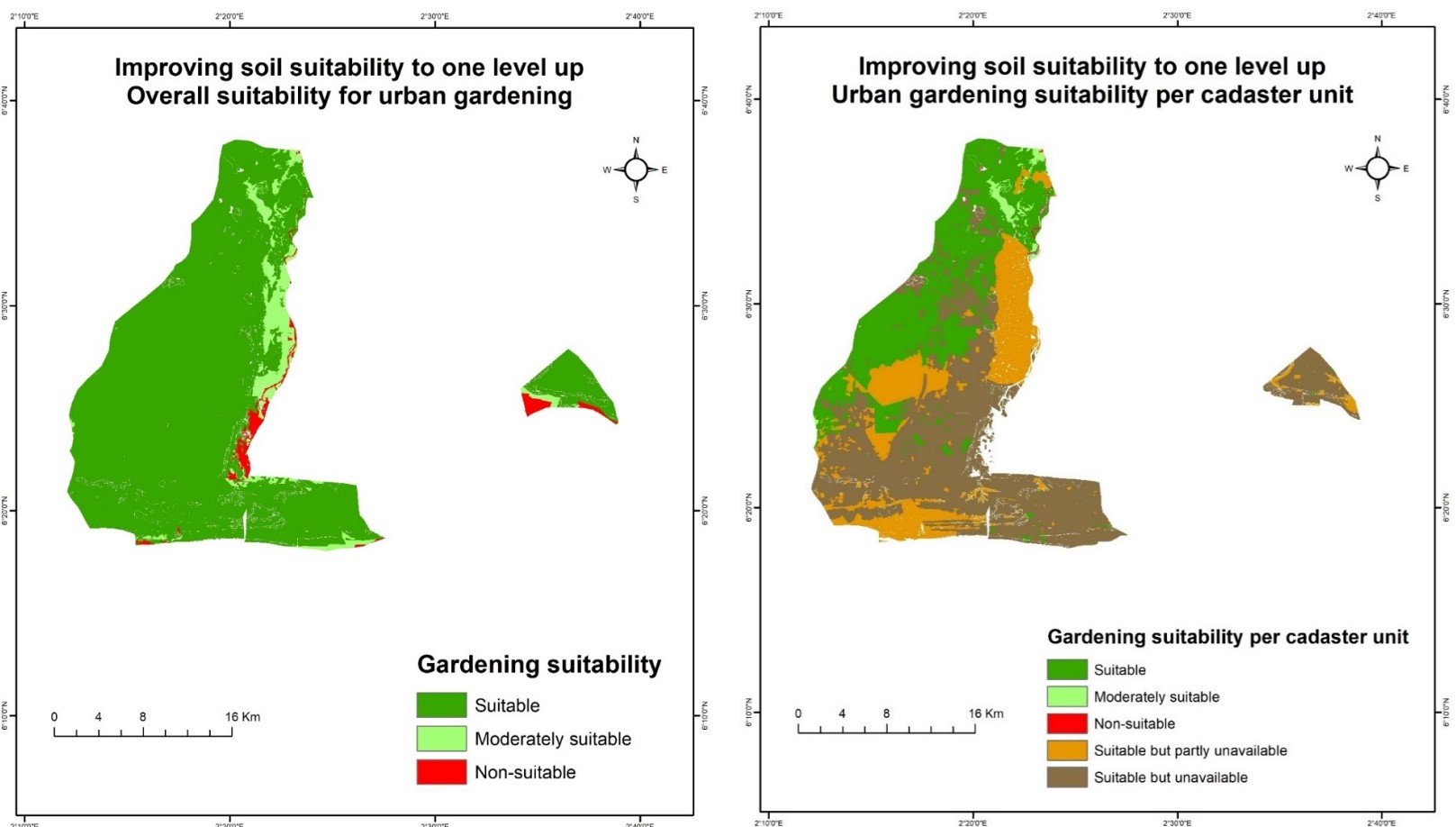

**Figure 4.** Classified maps of suitable areas for urban agriculture: soil improvement scenario.

0

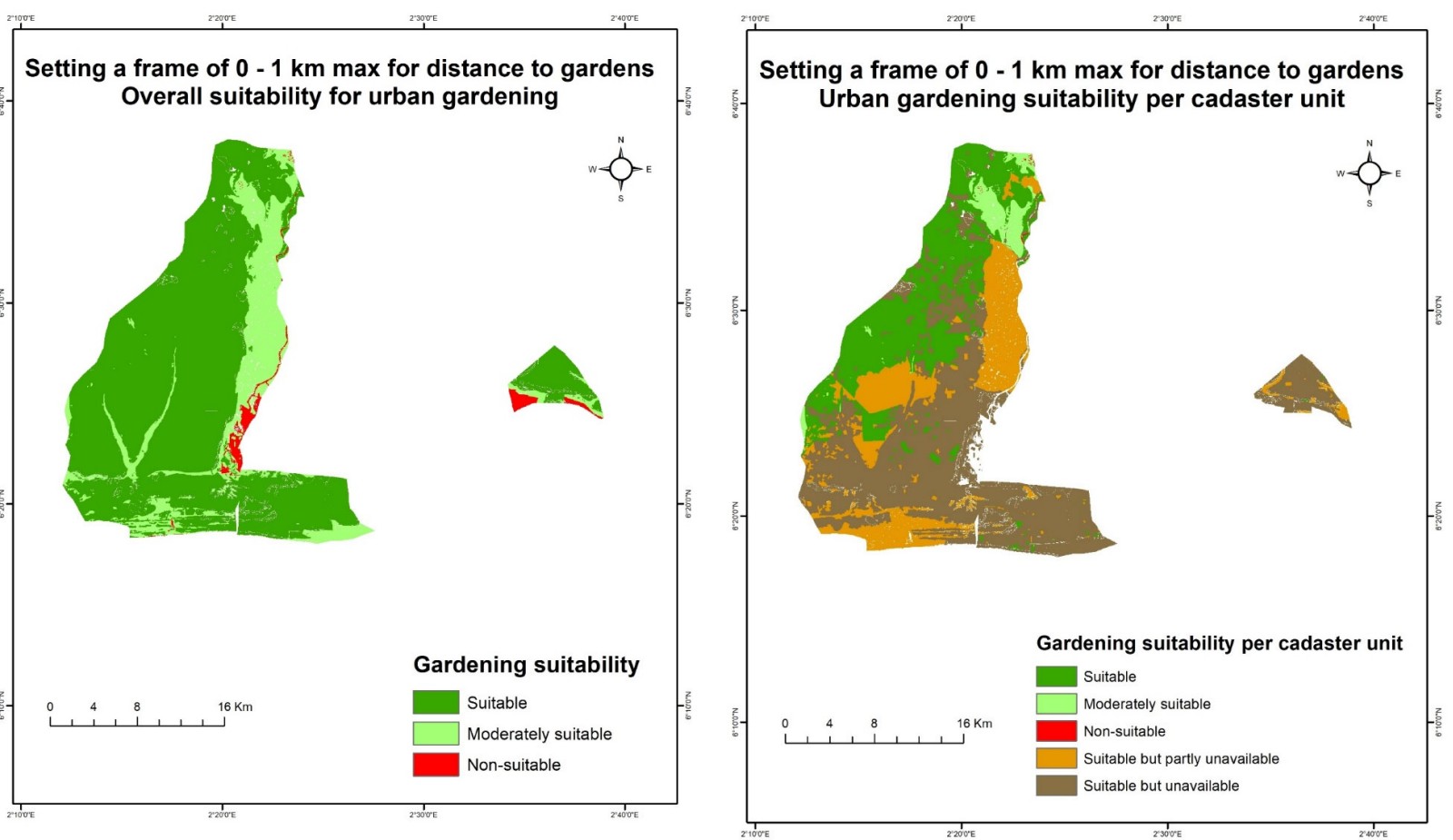

**Figure 5.** Classified maps of suitable areas for urban gardening: first safety improvement scenario.

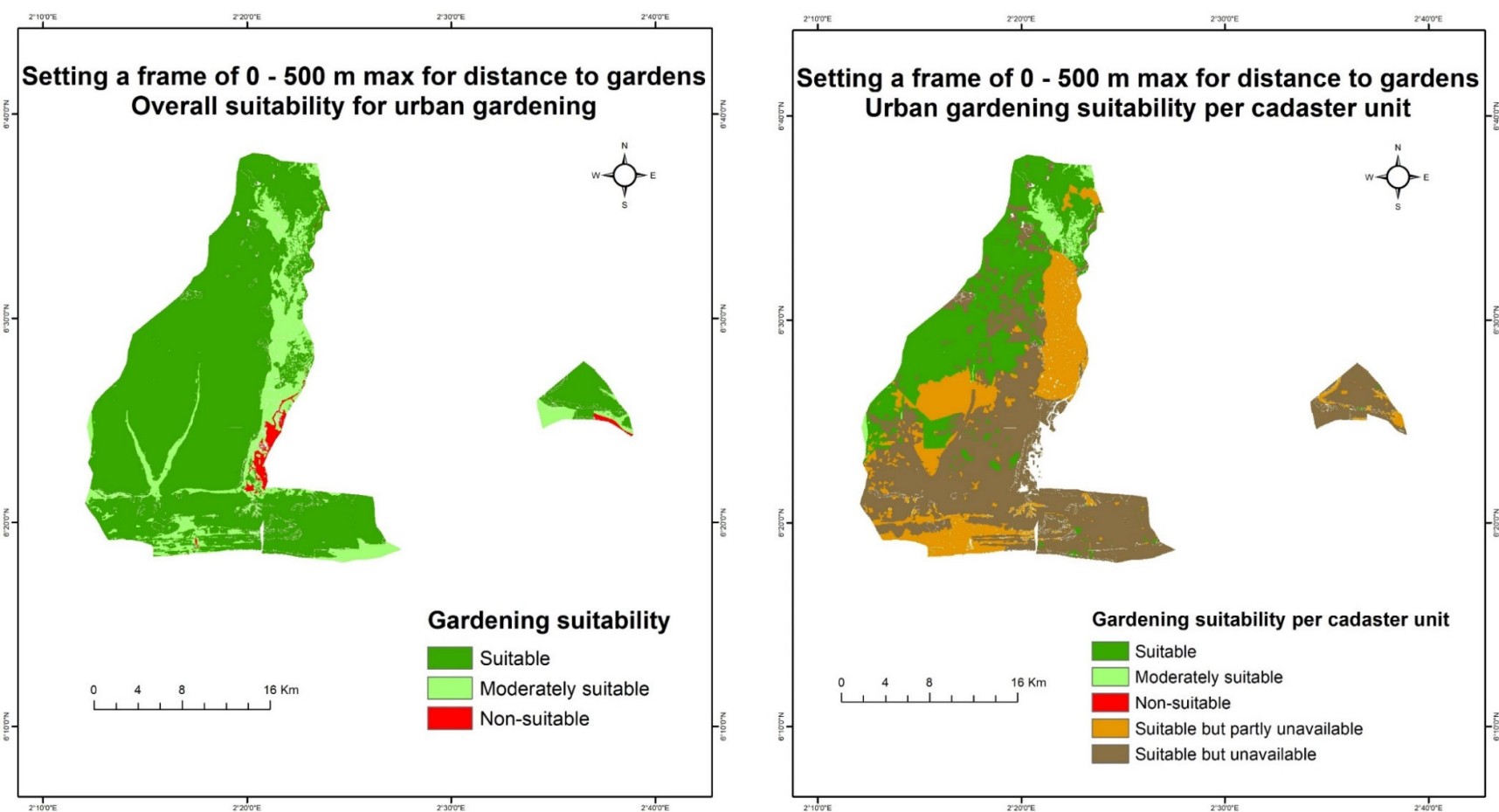

**Figure 6.** Classified maps of suitable areas for urban gardening: second safety improvement scenario.

## 3. Results

### 3.1. Overview of Data Used

Table 4 shows the areas for suitability classes of soil types, geology and land use and indicates the minimum and maximum values for road accessibility, safety for women, groundwater depth, length of growing period and population density. We observed for soil types and geology that 47 and 65% of the area was suitable or very suitable, respectively; 20% of the soil types and 18% of geology were unsuitable. For land use, almost the entire area is at least suitable. The distance to road and built-up area varied from 0 to 10 and 23 km, respectively. The digital elevation model showed that the land surface relief ranged from two meters under to 100 m below sea level. The length of growing period was between 242 and 260 days while the population density varied between 242 and 16,210 inhabitants per square kilometer.

**Table 4.** Description of independent variables.

| Variables | Soil Types | Geology | | Land Use | | |
| Suitability Levels | Area (km²) | Percent (%) | Area (km²) | Percent (%) | Area (km²) | Percent (%) |
|---|---|---|---|---|---|---|
| Unsuitable | 446.37 | 20.47 | 401.56 | 18.34 | 1.14 | 0.05 |
| Slightly suitable | 60.14 | 2.76 | 285.63 | 13.05 | 0.77 | 0.04 |
| Moderately suitable | 640.62 | 29.37 | 83.01 | 3.79 | 1.22 | 0.06 |
| Suitable | 918.92 | 42.13 | 822.57 | 37.57 | 744.81 | 35.25 |
| Very suitable | 114.93 | 5.27 | 596.39 | 27.24 | 1365.09 | 64.60 |

| Variables | Minimum | Maximum |
|---|---|---|
| Road accessibility (km) | 0 | 10.09 |
| Safety for women (km) | 0 | 22.60 |
| Groundwater depth (m) | −1.85 | 99.9 |
| Length of growing period (days) | 242 | 260 |
| Population density (inhabitants/km²) | 241.98 | 16210 |

### 3.2. Factors Contributing to Land Suitability for Urban Agriculture

The map of "Distribution of random points visited" on Figure 2 shows that the random points visited by experts to assess land suitability were well distributed over the study area. Table 5 summarizes the results of the stepwise ordered logistic regression and shows that five variables were significant (at the 5% level) in their contribution to the suitability of land for UA: soil suitability, land-use suitability, groundwater depth, safety for women and population density. The results showed that land suitable for UA is higher when soil and land-use suitability are improved. In addition, the odds of higher levels of land suitability are increased with higher population density, increased groundwater depth and short distance to a built-up area. Figure 2 depicted these five variables over the study area.

**Table 5.** Results of stepwise ordered logistic regression on urban gardening suitability.

| Parameter | DF | Estimate | Standard Error | Wald Chi-Square | Pr > ChiSq |
|---|---|---|---|---|---|
| Intercept 1 | 1 | −5.8226 | 1.5406 | 14.2836 | 0.0002 |
| Intercept 2 | 1 | −3.9468 | 1.5159 | 6.7787 | 0.0092 |
| Soil suitability | 1 | 0.5067 | 0.1871 | 7.3334 | 0.0068 |
| Land-use suitability | 1 | 0.8173 | 0.3362 | 5.9109 | 0.0150 |
| Groundwater depth | 1 | 0.0282 | 0.0114 | 6.1499 | 0.0131 |
| Safety for women | 1 | −56.5490 | 26.2683 | 4.6343 | 0.0313 |
| Population density | 1 | 0.000153 | 0.000058 | 6.9673 | 0.0083 |

The hit ratio and the number of correctly predicted classes that corresponded to the expert observations are shown in Table 6. The model correctly classified 63% of the observations. In 21 cases, the model underestimated suitability. For instance, in 16 cases where observations were suitable, the model predicted a moderately suitable (11) or unsuitable situation; when observations were moderately suitable, the model predicted in five cases an unsuitable situation. More seriously, the model overestimated suitability in 76 cases. For instance, in 46 cases where observations indicated that the land was moderately suitable, the model predicted a suitable situation; when observations indicated that the land was unsuitable, the model predicted in 30 cases a moderately suitable situation.

**Table 6.** Classified assessment of the expert model.

| Frequency % | Predicted Classes | | | |
|---|---|---|---|---|
| Observed Classes | Suitable | Moderately Suitable | Non-Suitable | Total |
| Suitable | 149 | 11 | 5 | 165 |
| | 56.23 | 4.15 | 1.89 | 62.26 |
| Moderately suitable | 46 | 14 | 5 | 65 |
| | 17.36 | 5.28 | 1.89 | 24.53 |
| Unsuitable | 0 | 30 | 5 | 35 |
| | 0.00 | 11.32 | 1.89 | 13.21 |
| Total | 195 | 55 | 15 | 265 |
| | 73.58 | 20.75 | 5.66 | 100.00 |

*3.3. Making Scenarios for Policy Interpretations*

The extrapolation of the model results to the entire research area in Figure 3 provided a baseline scenario of the overall suitability of land for UA. The flexibility of the tool was tested by observing changes in land suitability in two scenarios: improving soil suitability to one level up (Figure 4) and safety for women by reducing the distance to a built-up area (Figures 5 and 6). The results are summarized in Table 7 and show in the baseline scenario that 78 and 16% of the area corresponding to 416 and 84 km² were suitable and moderately suitable, respectively. By improving the soil suitability to one level up, a significant part of the area rose to higher suitability levels. For instance, there was an improvement of 13% corresponding to 67 km² of suitable land while the unsuitable area was reduced to almost 4% in the research area. The improvement of safety for women within a radius of one kilometer only slightly improved the situation; less than 1 and 3% improvement of a suitable and moderately suitable situation. The latter shows that the distance to a built-up area must be considerably closer to improve women's safety. By using 500 m, the changes were substantial in improving the land suitability; there was an improvement of about 4% (22 km²) of suitable area and a reduction of about 5% (25 km²) of unsuitable area.

**Table 7.** Urban agriculture suitability.

| Scenario Suitability | Baseline | | Soil Improved to One Level up | | Safety Improved in a Frame of 0–1 km | | Safety Improved in a Frame of 0–500 m | |
|---|---|---|---|---|---|---|---|---|
| | Area (km²) | Percent (%) | Area (km²) | Percent (%) | Area (km²) | Percent (%) | Area (km²) | Percent (%) |
| Suitable | 416.21 | 78.14 | 483.15 | 90.71 | 419.64 | 78.79 | 438.25 | 82.28 |
| Moderately suitable | 84.08 | 15.79 | 36.95 | 6.94 | 101.63 | 19.08 | 86.62 | 16.26 |
| Unsuitable | 32.33 | 6.07 | 12.53 | 2.35 | 11.36 | 2.13 | 7.76 | 1.46 |

By constraining the generated maps with land availability information from cadaster units, we created the overall suitability by cadaster unit for baseline (Figure 3), soil

improvement (Figure 4), and safety for women (Figures 5 and 6) scenarios. The purpose was to generate additional information for policies on the possibilities to develop UA. The results of changes observed are summarized in Table 8. We observed in general that the area suitable for UA was significantly reduced. In the baseline scenario, for example, the suitable and moderately suitable area was reduced to about 28 and 4%, respectively. There was, however, about 18% that may be partly used to develop UA. The same trend was observed in the scenarios for the improvement of soil and safety for women.

**Table 8.** Urban agriculture suitability combined with cadaster information.

| Scenario Suitability Levels with Cadaster Information | Baseline | | Soil Improved to One Level up | | Safety Improved in a Frame of 0–1 km | | Safety Improved in a Frame of 0–500 m | |
|---|---|---|---|---|---|---|---|---|
| | Area (km²) | Percent (%) | Area (km²) | Percent (%) | Area (km²) | Percent (%) | Area (km²) | Percent (%) |
| Suitable | 139.12 | 27.60 | 153.46 | 29.49 | 140.71 | 26.97 | 147.09 | 28.02 |
| Moderately suitable | 17.96 | 3.56 | 7.05 | 1.36 | 19.66 | 3.77 | 13.63 | 2.60 |
| Unsuitable | 3.70 | 0.73 | 0.27 | 0.05 | 0.41 | 0.08 | 0.06 | 0.01 |
| Suitable but partly unavailable | 91.86 | 18.23 | 106.2 | 20.41 | 107.11 | 20.53 | 107.50 | 20.48 |
| Suitable but unavailable | 251.36 | 49.87 | 253.39 | 48.69 | 253.78 | 48.65 | 256.64 | 48.89 |

## 4. Discussion

This study offers interesting insights into the integration of UA in land-use planning. Our research developed a site-selection tool that could support city authorities in making decisions about the allocation of land for UA. Specifically, the research successfully answered two research questions: how to estimate suitability for UA, and how to integrate the knowledge obtained into a site-selection tool.

### 4.1. Identifying Suitable Land for Urban Agriculture

We found five factors that significantly contributed to the suitability of land for UA: soil suitability, land-use suitability, groundwater depth, safety for women and population density. Soil fertility is an important factor and a starting point for UA. In a study of the densely populated Abbay region of Ethiopia using remote sensing, GIS, and an analytic hierarchy process to appraise land suitability for agriculture, soil suitability accounted for 61% of the overall land suitability [28]. The positive influence of soil quality on crop yield was also confirmed by Juhos et al. [38] in their study discussing the best multivariate statistical method in exploring the soil productivity function in an east Hungarian region. In addition, the type of land use hints at the possible development of UA, as described by Abebe and Megento [29] who found, in their study for urban green belt development in Addis Ababa, that land-use types are relevant indicators that contribute to the assessment of land suitability. Our study confirms this since we found that an area with water bodies or dense forests reduces suitability while an agglomeration or cultivated areas indicate a higher probability for developing UA. These findings are also corroborated by Yalew, van Griensven, Mul and van der Zaag [28] who found water bodies and forests as unsuitable for UA. We found that increased distance to the groundwater level improved the suitability of land, while we had assumed that suitable areas would be closer to groundwater, which increases access to a reliable source. Possible explanations are found in the low drainage capacity of the soils when groundwater is closer, and the greater likelihood of flooding. However, it is noteworthy that the highest altitude in our data domain was 100 m, which, may limit observations on the negative effects of high altitudes.

For instance, in their study, Abebe and Megento [29] found land lower and higher than 3700 m as suitable and unsuitable, respectively.

While statistics on thefts would be interesting to provide concrete information on safety for women, it was unfortunate that they were not available in a usable format. However, computing the Euclidean distance between grid cells and built-up area seems a reasonable proxy, i.e., that the closer the land is to a built-up area the safer women might be. Our results show that a greater distance reduces the likelihood of higher suitability for UA. Our finding was corroborated by Yalew, van Griensven, Mul and van der Zaag [28] who found that proximity to a town contributed to higher land suitability, though they did not discuss it from a gender perspective. Therefore, our findings add value to the body of knowledge identifying women as a vulnerable group to be considered in order to ensure that planned interventions are gender sensitive. Taiwo [39], in his study on the area chosen for UA among urban farmers in the Nigerian city of Lagos, discussed constraints related to greater distance from people's place of residence from two perspectives. First, the greater the average physical distance, the higher the cost of transport, reducing profitability. Second, a greater distance may increase commuting time, and so reduce the time spent on agricultural activities. Hence, we can conclude that distance is an important indicator that requires special attention, especially from a gender perspective, in the planning for policy interventions regarding UA. Furthermore, we found that higher population densities improved land suitability. The explanation for this result is simple: a higher population density means a larger market and a greater likelihood of being able to sell highly perishable surplus produce. Hence, our findings imply that when markets are closer, land suitability is higher. A similar finding is made by Taiwo [39], namely a preference for farms that minimize the overall cost of transporting produce to markets.

Moreover, we expected road accessibility and geological suitability to be significant in the model. Indeed, the findings of Yalew, van Griensven, Mul and van der Zaag [28] found a shorter distance to roads as an important factor for higher land suitability. Hence, we assumed that the effect of road accessibility in our study might have been hidden by the distance to a built-up area (used as a proxy of safety for women) because it is highly likely that densely populated areas will be close to roads. Geological suitability is also a good indicator of the quality of soil types present in an area. We assume that its non-significance might be explained from two perspectives: either its effect was hidden by soil suitability or its diversity in our study area was too small to have a significant effect on land suitability. Therefore, we recommend that future research over a broader area considers the two variables for the further exploration of their contribution to land suitability. In addition, although water accessibility (distance to water bodies) was omitted in our study due to unavailable data, it could be considered in future research because water is an essential requirement that easily guarantees the possibility of UA in any location [28,39,40].

### 4.2. Integrating Acquired Knowledge into a Tool for Land Allocation

Our research attempted to integrate the knowledge obtained into a site-selection tool that could help policymakers to allocate land to the development of UA. Indeed, the above-described criteria were combined to generate comprehensible maps that indicated suitable areas in the cities examined. More importantly, the capacity of the tool is underlined by indicating the changes in land suitability that might be created by various policy interventions. For instance, the improvement of soil suitability (to one level up) and safety for women (within a distance frame of 500 m) significantly increased the suitable area (13 and 4% more, respectively) within cities and the unsuitable space was reduced. The tool's dynamic capacity makes it appropriate for use in various contexts and for scaling up policies that seek to achieve more impact in supporting the reduction in poverty in peri-urban and urban areas. In addition, the dynamic feature of the tool makes it possible to add new variables that might be important in contributing to land suitability.

For instance, the previously mentioned challenge of the lack of accessible information made it difficult to examine the diversity and wealth of potential information that might improve land suitability. The challenge of the accessibility of information affects both access to available data and the production of new data that provide accurate information on the reality on the ground. Therefore, it is crucial that local and national planning policies produce and make available accurate data regarding various indicators of agricultural potential to develop better context-specific solutions to emerging issues such as the rapid and unplanned urbanization of Benin in particular and African countries in general. In this COVID-19 era, the tool could also be applied to evaluate the areas of the "cordon sanitaire" in Benin, by considering access to UA as a legitimate exemption from the lockdown restrictions. A study [41] shows the negative impact on food security when access to UA allotments is restricted.

Furthermore, our research made scientific and social innovations. First, we adopted a transdisciplinary approach that combines academic and non-academic (experts) knowledge to generate a new knowledge base that best integrates scientific and practitioner perspectives to propose field-informed solutions to urban planning policies. The approach surveyed the literature, discussed potential variables of land suitability with practitioners and produced a set of relevant criteria that formed the basis of the study. The field observations made during randomly determined site visits were combined with standardized spatial and non-spatial datasets using statistic and estimation procedures to generate a robust model that serves as an input for extrapolation to the entire area, making it dynamic for weighted regressions that should account for various policy intervention choices. The approach, therefore, goes beyond the usual static techniques that use GIS-based multi-criteria analysis and an analytical hierarchy process for weighted regressions to assess land suitability and verify it subsequently, as shown for example in many studies on agriculture [28,42], watershed management [30,43], forest growth [44,45], urban green space [29], and urban development [27,46]. Then, our study generated new and easy-to-use information that should contribute to the formulation of urban policy interventions aiming to support poor people in need of land to produce fresh foods to reduce their vulnerability to food insecurity and hunger.

## 5. Conclusions

Our study addressed calls from African countries that aim to develop policies to allocate land to UA and support the reduction in food insecurity and poverty in urban areas. Learning from the case of Benin, our research identified suitable land for UA and extrapolated the results into a land-allocation tool that could support decision-making processes about the integration of UA in urban land-use planning. The tool would help demarcate suitable land within peri-urban and urban areas and further advise on suitable land that is available for use because, contrary to the common belief, even in highly urbanized areas, it is possible to find plentiful vacant land that can be used for agriculture on a temporary (land reserved for other uses) or permanent basis. The capacity of the tool also comes into its own by showing the changes made on suitable areas by various policy measures (improvement of soil fertility, and safety for women) that could improve the suitability of available land. However, identifying a suitable area is not enough to enhance the access of poor people to land. There is a need for strong political will at the national and local level, accompanied by clearly defined mandates and guidelines on land allocation. Furthermore, policymakers could formally integrate UA into urban development plans and policies addressing poverty and reducing food insecurity.

**Author Contributions:** Conceptualization, B.G.J.S.S., M.D.H. and A.A.; methodology, B.G.J.S.S., M.D.H. and G.J.M.v.d.B.; software, B.G.J.S.S., M.D.H. and G.J.M.v.d.B.; validation, B.G.J.S.S.; formal analysis, B.G.J.S.S. and M.D.H.; investigation, M.D.H. and A.A.; data curation, B.G.J.S.S., M.D.H. and G.J.M.v.d.B.; writing—original draft preparation, B.G.J.S.S. and M.D.H.; writing—review and editing, G.J.M.v.d.B. and A.A.; supervision, B.G.J.S.S.; funding acquisition, M.D.H. and B.G.J.S.S. All authors have read and agreed to the published version of the manuscript.

**Funding:** This research was funded by the Dutch Research Council (Dutch acronym: NWO), File no: 08.260.302 and the Organisation for Internationalisation in Education in the Netherlands (Dutch acronym: Nuffic): no. R/003248.01.

**Institutional Review Board Statement:** Not applicable.

**Informed Consent Statement:** Not applicable.

**Data Availability Statement:** The data regarding the explanatory variables are taken from publicly available datasets, see Table 1. The survey data for the dependent variable are available on request from the corresponding author.

**Conflicts of Interest:** The authors declare no conflict of interest.

## Appendix A

**Table A1.** Evaluation of soil, geological and land-use suitability (original French classification).

| Items | Suitability Level |
|---|---|
| **Soil types** | |
| *Plan d'eau* | Unsuitable |
| *Sols peu évolués hydromorphes* | Slightly suitable |
| *Vertisols hydromorphes* | Slightly suitable |
| *Sols ferrugineux tropicaux lessivés sans concrétions* | Moderately suitable |
| *Sols ferrugineux tropicaux lessivés hydromorphes* | Slightly suitable |
| *Sols ferrallitiques faiblement désaturés appauvris modaux* | Suitable |
| *Sols ferrallitiques faiblement désaturés appauvris hydromorphes* | Moderately suitable |
| *Sols hydromorphes moyennement organiques humiques à gley* | Moderately suitable |
| *Sols hydromorphes minéraux ou peu humifères à gley de profondeur* | Moderately suitable |
| *Sols hydromorphes minéraux ou peu humifères à gley lessivés* | Suitable |
| *Sols hydromorphes minéraux ou peu humifères à pseudo-gley* | Very suitable |
| **Geology** | |
| *Plan d'eau* | Unsuitable |
| *Quaternaire: sables éoliens et marins du cordon littoral actuel et récents, alluvions argilo-sableuses des vallées des intérieures des fleuves* | Moderately suitable |
| *Quaternaire: alluvions argilo-sableuses des vallées intérieures des fleuves* | Very suitable |
| *Quaternaire: terrasse de 5 à 40m très développée en argile et sable sur la façade littorale* | Slightly suitable |
| *Continental terminal mio-pliocène: argile rouge latéritique, argile bariolée, argile sableuse, noire ou colorée, sable fin à grossier, grès.* | Suitable |
| *Continental terminal éocène et paléocène: argile gris-bleu, berge-galet de quartz, sable fin blanc, calcaire phosphate.* | Moderately suitable |
| *Cambrien protérozoïque supérieur: argilites, sils, grès fin, quartzites fins et moyens, siltites, jaspes, schistes argileux.* | Suitable |
| *Panafricain: syéno-monzonite, granites microsyénites, granites calco-alcalin syntectoniques, charnockites, gness granitoides, granites nigmatiques* | Very suitable |
| **Land cover** | |
| *Forêt galerie* | Slightly suitable |
| *Forêt dense* | Unsuitable |
| *Forêt marécageuse* | Unsuitable |
| *Forêt claire et savane boisée* | Slightly suitable |
| *Savane arborée et arbustive* | Slightly suitable |
| *Plantation* | Moderately suitable |
| *Cultures et jachères* | Very suitable |
| *Cultures et jachères à palmiers* | Suitable |
| *Riziculture* | Slightly suitable |
| *Mangrove* | Unsuitable |

| *Marécage* | Unsuitable |
| --- | --- |
| *Plan d'eau* | Unsuitable |
| *Agglomération* | Very suitable |
| *Sol nu* | Unsuitable |

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
