# Peer review of "Where Do I Allocate My Urban Allotment Gardens? Development of a Site Selection Tool for Three Cities in Benin"

_land, doi:10.3390/land10030318_

Round 1

Reviewer 1 Report

Dear Author,

The manuscript " Where do I allocate my urban allotment gardens? Development of a site selection tool for three cities in Benin. " describes very interesting issues. The discussed topic is extremely timely, important and necessary.

The reviewer has several comments and suggestions, the inclusion of which will improve the quality of the manuscript.

The authors should describe the methodology in more detail. The description of the regression method is insufficient. This does not allow the reader to closely follow the course of the proceedings. At the same time, the literature review is modest and it does not help to fill the gaps in the method description. The literature review should be more detailed.

Another less significant note concerns manuscript formatting. First of all, the maps are too small. You have a lot of empty space around and the map is tiny and unreadable. Increase the scale.

The text is also badly formatted. Always check the pdf before uploading the article to the system. It is insignificant but it discourages the reviewer.

The literature list is also badly formatted. Correct it.

Apart from these comments, I believe that the assumptions and goals are correct and the conclusions are correctly constructed.

Reviewer 2 Report

The study is approaching an interesting topic and brings new insights into urban agriculture. Still, some changes should be done in order to make the research easier to follow and for a better understanding of certain ideas. 

In the introduction should be mentioned the state of the art regarding the ways of suitability assessment for urban agriculture.

Section 2 (Materials and Methods) should contain a relevant description of the three case-studies as the section 2.1. Study area. Also, should outline the reason for choosing these 3 cities.

More information is needed in section 2.1. (2.2. after introducing Study area) Compilation of independent and dependent variables

Line 135-138. During a three-day workshop with UA experts and researchers a set of criteria was defined in relation to the suitability for allotments. In a first attempt, a full list of independent variables was prepared and discussed in plenaries.

Please explain how many experts were involved, and which is their profile? How many researchers were involved and from what field of research? Which were the criteria based on which you prepared a full list of independent variables?

Line 157-159. We compiled the dependent variable on the suitability of sites for UA on the basis of assessments by five UA experts who visited and assessed 60 randomly selected sites in our research area. 

This phrase is somehow unclear to me. They actually visited the sites? If the analysis was done using different databases what was the purpose of these visits? How did you choose the five experts?

Figure 1, Figure 2, and Figure 3. It cannot be distinguished from these figures where are located the three cities. Please, mark their administrative boundaries and names on the maps.

Figure 1 Population density map. On the legend, please round the number – e.g. 241.975 will be 242 and so on.   

Figure 1 Distribution of random points visited. What represents the background colors on the map?

Table 3 – should be translated into English

The expression land geology is not used properly, the right term is geology, considering that geology and land are two different fields. Please modify all over the paper.

The differences between the three case studies should be highlighted and explained in the results section and also in the discussion section.

Reviewer 3 Report

The paper focuses on the development of a decision-making support tool that assesses the suitability of land for allotment gardens in urban and peri-urban areas.
The paper is in general well-constructed and the topic is relevant, especially in this COVID-19 era, as the Authors sustain, in which access to UA could be considered a legitimate exemption from the lockdown restrictions.
Moreover, the methodology is clearly expressed, and this increases the possibility of its reproduction even with poor data available.

I found some criticalities, especially related to the use of cartographic apparatus.
Below I report (in summary) the main notes.

  • In general, a better link between graphic material and the text is suggested. Figure 1 for instance is inserted (lines 175-222) quite far from the text mentioning it (line 301-311). This reduces the general readability of the text. Moreover, it is suggested to insert a first map showing the localization of the selected area in a more vast geographic context.
  • It can be useful to insert in Section 2 a diagram graphically reproducing the different methodological steps followed by the study. This is important to compare this study with other techniques that use GIS-based multi-criteria analysis.
  • Few minor notes:
    - Line 281, please indicate the figure you are referring to
    - Line 301, please explain the reason why Figure 1 shows that the random points visited by experts to assess 301 land suitability were well distributed over the study area.
    - Line 525, please mention few policy measures that could improve the suitability of available land. This could improve the value of the conclusions.

Round 2

Reviewer 2 Report

The Authors have properly addressed all my concerns. Thus, the paper can be accepted in the present form.

Author Response

Dear Madam, Sir, our response can be found in the attached file.

Kind regards,

Ben Sonneveld
